# Hydration, Arginine Vasopressin, and Glucoregulatory Health in Humans: A Critical Perspective

**DOI:** 10.3390/nu11061201

**Published:** 2019-05-28

**Authors:** Harriet A. Carroll, Lewis J. James

**Affiliations:** 1Department for Health, University of Bath, Claverton Down, Bath BA2 7AY, UK; 2Rowett Institute, University of Aberdeen, Ashgrove Road West, Aberdeen AB25 2ZD, UK; 3Exercise and Health Sciences, School of Sport, Loughborough University, Epinal Way, Loughborough LE11 3TU, UK; L.James@lboro.ac.uk

**Keywords:** vasopressin, copeptin, hydration, health, metabolism, glycaemia, type 2 diabetes, diabetes insipidus, syndrome of inappropriate anti-diuretic secretion, MDMA

## Abstract

Glucoregulatory diseases, such as type 2 diabetes are currently a key public health priority. Public health messages have started to include the addition of water in their dietary guidelines. Such guidelines however are not based on causal evidence pertaining to the health effects of increased water intake, but rather more heavily based upon non-causal or mechanistic data. One line of thinking linking fluid intake and health is that hypohydration induces elevated blood concentrations of arginine vasopressin (AVP). Research in the 1970s and 1980s implicated AVP in glucoregulation, supported by observational evidence. This important area of research subsequently appeared to stop until the 21st century during which interest in hypertonic saline infusion studies, animal AVP receptor knockout models, dietary and genetic associations, and human interventions manipulating hydration status have resurged. This narrative review briefly describes and critically evaluates the usefulness of the current AVP-glucoregulatory research. We offer suggestions on how to test the independent glucoregulatory effects of body water changes compared to elevated circulating AVP concentrations, such as investigating hydration manipulations using 3,4-Methylenedioxymethamphetamine. Whilst much research is still needed before making firm conclusions, the current evidence suggests that although AVP may be partially implicated in glucoregulation, more ecologically valid models using human participants suggests this effect might be independent of the hydration status. The key implication of this hypothesis if confirmed in future research is that manipulating the hydration status to reduce circulating AVP concentrations may not be an effective method to improve glucoregulatory health.

## 1. Introduction

Historically, research in hydration focused on large deviations in hydration status. Specifically, the ill-effects of severe hypohydration in soldiers under extreme conditions were investigated, resulting in guidelines for optimal sports performance [1]. Following this, public health guidelines started to incorporate hydration recommendations; a more detailed description of this shift has been reviewed previously [1]. Briefly, such recommendations may have oversimplified the complex relationship between fluid intake and health. One of the most prominent examples is the Institute of Medicine guidelines which noted that serum osmolality stays within a well-defined range across a multitude of fluid intakes; subsequent guidelines were therefore based on median intakes of self-reported fluid ingestion (‘adequate intake’) [2]. Thus, to date, fluid intake guidelines have not been based on evidence pertaining to improved health, such as lower risk of glucoregulatory diseases (e.g., type 2 diabetes [T2D]).

Understanding the true causal role of hydration status in health and disease is important in order that guidelines are based on the best possible evidence. Increasing fluid intake (particularly from water) as a means to manipulate hydration status (and therefore potentially health) represents a low cost and easy to understand intervention. Further, the addition of water to the diet does not remove hedonically rewarding foods or beverages, the removal of which may contribute to poor adherence when implementing a dietary intervention, though more research is certainly warranted to understand adherence to recommendations surrounding increasing fluid intake.

One key mechanism linking hydration status to glucoregulatory health is arginine vasopressin (AVP) which is a hormone implicated in body water regulation. This hormone is typically known for its impacts on blood pressure regulation, whereby hypohydration (as detected by a 1–2% increase in serum osmolality) is met by an increase in circulating AVP [3]. The result of this is V2 receptor binding in the collecting ducts of the kidney, signalling an increase in aquaporin expression and redistribution to the luminal membrane [4]. This increases water reabsorption, attenuating a reduction in blood volume when water intake is low [5]. However, it is important to acknowledge that many factors can alter circulating AVP concentrations, including genetics [6], ambient temperature [7], circadian rhythms [8], pharmaceuticals [9], recent fluid intake [10], and stress [11]. Thus, the pathway from low fluid intake to high circulating AVP to poor glucoregulatory health is difficult to examine as the high AVP may be due to other extraneous variables.

Nonetheless, high plasma concentrations of AVP have been associated with poorer cardiometabolic and glucoregulatory health [12,13]. Water intake almost immediately reduces plasma copeptin concentrations (a surrogate marker of AVP [14]) for >4 h. This has led some to hypothesise that interventions to reduce copeptin via increasing fluid intake may facilitate positive health outcomes (e.g., [1,10]). This line of thinking may, however, oversimplify the relationship between hydration, AVP, and health. The aim of this narrative review is to briefly discuss early and current human research in hydration, AVP, and glucoregulatory health and provide a critical perspective as to how to advance the field, focusing on uncoupling the effects of hydration status and AVP.

## 2. History

A summary of the studies discussed herein can be found in Table 1. Much research investigating the role of AVP in water balance was conducted in the 1970s (e.g., [15,16]) subsequently resulting in data suggesting AVP may be linked to glucoregulation. In 1979, Zerbe et al. [17] found elevated plasma AVP concentrations in patients with uncontrolled diabetes mellitus (i.e., hyperglycaemia). This seemed counterintuitive as hyperglycaemia is typically accompanied by polyuria, which acts to help maintain euglycaemia. Rather, these patients had severe hypovolemia sufficient enough to induce AVP secretion. It therefore appeared that hyperglycaemia caused polyuria, resulting in hypovolemia [17]. Consequently, plasma AVP concentrations increased to counteract the loss in blood volume, which may have been potentiated by the osmotic effect of high blood glucose concentrations [17]. When treated with insulin (and hydrated), plasma AVP concentration reduced by up to five-fold, along with concomitant reductions in plasma osmolality [17]. Such findings were key in introducing the idea that AVP may have a role in glucoregulation. 

Following this, in 1985, Spruce et al. [22] advanced this research, along with theory from research in rodents and dogs (e.g., [28,29]), by infusing AVP into healthy adults and measuring glucoregulation (including glucose kinetics using labelled glucose). Plasma concentrations of AVP reached 22.3 ± 5.4 pmol∙L^−1^ during a 30 min low dose infusion (25 pmol∙min^−1^), and 112.3 ± 18.4 pmol∙L^−1^ during a subsequent 60 min high dose infusion (75 pmol∙min^−1^), without altering plasma osmolality. Arterialised-venous blood glucose concentrations increased from 4.9 ± 0.1 mmol∙L^−1^ to 5.2 ± 0.2 mmol∙L^−1^ after the low dose infusion and to 5.7 ± 0.2 mmol∙L^−1^ after the high dose. Such changes were not found after saline infusion (though no details were given regarding the saline so it is assumed that this was isotonic). No effects from any treatment were found for plasma insulin concentrations, though plasma glucagon concentrations were ~41 pg∙L^−1^ higher during the low dose AVP infusion, which remained throughout the high dose infusion.

Such studies offered insights into potential mechanisms by which AVP might be implicated in glucoregulatory health. Firstly, as can be seen from the work of Zerbe et al. [17], there is a complex interplay between hyperglycaemia and AVP, potentially mediated by hypovolemia induced by glucosuria. Although the study was unable to determine the temporal direction of the hyperglycaemia-AVP relationship, considering the elevated AVP was accounted for by hypovolemia, it is likely that the hyperglycaemia drove higher AVP, rather than vice versa (Figure 1). The AVP-induced hyperglycaemia may have also created an osmotic stimulus further stimulating AVP secretion (Figure 1).

Secondly, Spruce et al. [22] demonstrated an increase in glucose production but not disposal at supraphysiological circulating AVP concentrations; this was likely driven by glycogenolysis, rather than gluconeogenesis, as there were no changes in gluconeogenic precursors such as lactate. The increase in glycaemia was therefore likely due to the greater plasma glucagon concentrations during AVP infusion. The divergence in mechanistic findings of these two studies provides a strong rationale to separate patients with diseases from healthy participants when making comparisons between theories and studies.

Beyond the direct role of AVP on glucoregulation, theoretical implications have also been hypothesised involving the hypothalamic-pituitary-adrenal (HPA) axis, which were highlighted in the 1980s. The HPA axis is implicated in the stress response which can increase hepatic glucose output. Upon experiencing stress, AVP and corticotropin-releasing factor (CRF) are synthesised and secreted by the hypothalamus. These hormones regulate adrenocorticotropic hormone (ACTH) secretion from the pituitary gland, resulting in cortisol secretion from the adrenal cortex. In the 1980s, there was sufficient evidence, primarily from animal models, regarding the role of AVP in this response [30]. Specifically, during physical stress, CRF (a precursor to ACTH) is produced. In vivo, AVP can potentiate the effect of CRF on ACTH production. As AVP is elevated during physical stress, there remains a clear pathway between AVP and the stress response, via CRF, ACTH, and ultimately cortisol secretion [30]. Accordingly, it may be that increasing water intake to reduce AVP could mitigate excessive cortisol secretion, thus reducing hepatic glucose output (Figure 2).

## 3. Current Research

As discussed in the previous section, the studies conducted in the 1970s and 1980s provided fascinating insights into the role of hydration and AVP in glucoregulatory health. Yet, despite the broad ranging implications of the work, interest in AVP waned until the 2000s. Recently, research has expanded the early work and helped provide some critical perspective.

Water intake can be used as a crude proxy for hydration status. Few studies have investigated the relationship between water intake and markers of glucoregulatory health. In a French cohort of men and women, higher water intake (assessed via self-reported categories of litres of plain water per day) was associated with lower risk of hyperglycaemia [18]. This relationship was replicated in a small UK sample of men and women [20] but not in a sample of US female nurses [19] (both of which assessed water intake using a food frequency questionnaire). Following these studies, a representative UK sample also found an inverse relationship between plain water intake (from four-day unweighed diet diaries) and glucoregulatory health, though upon further analysis, this association was only found in men [21]. This latest study may explain the null relationship found in US nurses [19] due to the sample being exclusively women, compared to the other studies which used mixed sex samples. Regarding AVP, this could make sense due to fluctuations in the osmolality set-point for AVP secretion during the menstrual cycle [31], which may cloud any associations. Alternatively, the unvalidated methods of fluid intake may not have accurately captured true water intake, explaining the mixed findings.

Further observational evidence has more directly implicated AVP in glucoregulatory health. In a Swedish sample, higher plasma copeptin (as a surrogate marker of AVP) concentrations were associated with higher prevalence of T2D and insulin resistance, both cross-sectionally and longitudinally [12]. Advancing these associations are studies investigating variations in AVP receptor genes. Participants with the variation in the AVP1a receptor gene (specifically, rs1042615 T allele) had a higher prevalence of T2D in those with a high fat diet or with overweight [32]; similarly AVP1b receptor genes were tentatively associated with increased T2D risk [33].

Of course, observational evidence has limited causal inference due to well-known problems such as reverse causality and residual confounding. Particularly in the case of water intake, beyond issues surrounding misreporting, water intake is part of a cluster of other healthful behaviours such as higher fibre intake [34], which may confound any associations. In the case of the genetic observations, these could be chance associations, or AVP receptor genes may be collinear with other genes, which directly cause disease, thus representing a marker of a different mechanistic process. Nonetheless, the genetic work shows some agreement with rodent models, which have used knockout models, improving the confidence in these findings [35]. Alternatively, as the outcome for these studies was related to glucoregulatory health, there may have been confounding from unmeasured hypovolemia, in line with the work of Zerbe et al. in those with uncontrolled diabetes mellitus [17]. In other words, the findings may not be due to changes in hydration status or AVP *per se*, but rather represent a response (symptom) to poor glucoregulatory health resulting in glucosuria; AVP then responded to maintain blood volume in response to excessive urinary water losses.

Following these studies, interest has started to focus on understanding the causal relationship between hydration status, AVP, and glucoregulation. One study has investigated the acute and medium-term effects of water ingestion on glucoregulation. In this study, participants consumed one litre of water, after which their copeptin reduced within 30 min and remained suppressed by ~39% throughout the full test period of 4 h [10]. Furthermore, after one week of increased water ingestion, copeptin was reduced by 15% compared to a control week [10]. This study further split participants into ‘responders’ (i.e., copeptin reduced significantly after water ingestion; typically those with habitually low water intake, high copeptin and elevated urine osmolality) and ‘non-responders’ (i.e., water ingestion did not meaningfully impact their copeptin; typically those with habitually higher water intakes and low copeptin) [10]. In ‘responders’, increasing water intake did not result in changes in plasma glucose or insulin concentrations, but glucagon concentrations did reduce. Such work builds on that of the AVP infusion study [22], by demonstrating a reverse effect. In other words, AVP infusion [22] induced higher glucagon concentrations, whereas water intake prescription [10] induced lower glucagon concentrations by reducing AVP concentrations (via increasing fluid intake). Taken together, there appears to be a dose-dependent effect of AVP on circulating glucagon concentrations with mixed findings on whether this impacts glycaemia.

Arginine vasopressin secretion can be induced by small increases in serum osmolality [15]. Accordingly, some studies have investigated the role of hypertonic saline infusion on glucoregulation. After hypertonic saline and desmopressin infusion plus fluid restriction in healthy adults, Keller et al. [23] found higher fasted plasma glucose concentrations, coupled with an increase in endogenous glucose appearance, compared to an iso- and hypo-osmotic trial arm. Similarly, hypertonic saline infusion before an oral glucose tolerance test (OGTT) resulted in higher postprandial glucose concentrations 60 and 90 min post-glucose ingestion [24]. It is unclear from these studies whether the effect seen was directly due to the saline, or indirectly due to hyperosmolality-induced AVP secretion (or another as yet unknown mechanism).

In terms of the glucoregulatory impacts of direct manipulations to hydration status, there is limited current evidence and no replication studies. In 2001, Burge et al. [25] withdrew insulin in male and female patients (*n* = 10) with type 1 diabetes (T1D) and hypohydrated them via fluid restriction (750 mL per 24 h) and both oral and intravenous diuretics, losing on average ~4.1% of their body mass compared to a euhydrated control arm. Eight h after a set meal, insulin was withdrawn and biochemistry was measured for five h. Hypohydration resulted in an elevated glycaemic response compared to the control arm [25]. In a subgroup of participants, hypohydration was found to result in lower glucosuria compared to the control arm by an amount concordant with the difference in glycaemia between the two trial arms. Beyond glucosuria, compared to the euhydrated control arm, hypohydration induced higher plasma glucagon and cortisol concentrations, which may also explain the higher glycaemia found.

Similarly, Johnson et al. [27] hypohydrated medication-withdrawn men (*n* = 9) with T2D (~1.6% body mass loss) via fluid restriction, reporting elevated postprandial serum glucose concentrations with hypohydration. No differences in plasma insulin were found between trial arms. Mechanistically, there was no difference in the renin-angiotensin-aldosterone system (RAAS) according to hydration status, but the authors were unfortunately unable to measure AVP. Nonetheless, plasma cortisol concentrations were lower 45 min post-glucose ingestion during the euhydration trial arm [27]. However, whilst there were main effects and interaction effects, no post-hoc differences were found between the trial arms. Additionally, the time course of change in cortisol concentration does not clearly correspond to that of the glycaemic response. Thus, it may be that there is an interaction between hydration status and nutritional status, which mediated a cortisol response, rather than cortisol playing a role in hydration-induced alterations in glucoregulation. Alternatively, the cortisol trend may be a response to the medication withdrawal, since it is similar (though with a more rapid onset) to the work in insulin-withdrawn participants with T1D [25]. Unfortunately, the authors did not measure glucosuria for comparison with those with T1D [25].

Considering that early work suggested differential mechanisms between those with glucose dysregulation [17] and those who are healthy [22], we conducted a pilot study in healthy adults. In this study, we hypohydrated participants using a sauna and fluid restriction protocol and subsequently conducted an OGTT [26]. As per the research in people with diabetes, we also found a higher glycaemic response during hypohydration compared to euhydration. The difference in glycaemia emerged after 30 min (slightly earlier than those with T2D [27]). As per the AVP-infusion study by Spruce et al. [22] we also did not find a difference in lactate in our pilot work [26], though it tended to be higher during hypohydration 60 min post-glucose ingestion. However, this pilot study lacked rigorous control (e.g., verbal compliance only for the 24 h pre-trial standardisation), had a small sample (*n* = 5), and was unable to measure mechanisms. The consistency and clarity in the results seemed somewhat incredible, thus warranting a tightly controlled follow-up study.

Therefore, our follow-up study used much more rigorous pre-trial standardisation (four days of food, fluid, and physical activity replication) and measured a multitude of mechanisms [11]. In this study, participants lost ~1.9% body mass during hypohydration, serum osmolality increased by ~9 mOsm·kg^−1^, and their plasma copeptin concentrations increased from levels typically seen in healthy adults to levels seen in those with diabetic ketoacidosis [36]. Thus, we are confident that we induced meaningful changes in both water balance and AVP concentrations.

Despite this, we did not find a difference between trial arms in the arterialised-venous serum concentrations of glucose or insulin (neither fasted, nor postprandial) [11]. At 45 and 60 min post-glucose ingestion, there was a small divergence between the trials, similar to our pilot study [26], though these data were non-significant and non-meaningful (unlike our pilot). Importantly, there were no differences in fasting or postprandial ACTH or cortisol concentrations, contrary to participants with T1D [25] and T2D [27]. This may suggest that the interaction between hydration status and cortisol responds differentially during medication withdrawal in diabetes compared to healthy adults. Additionally, as our results in healthy adults are divergent to those in diabetes, it seems likely that glucosuria during euhydration provides a better explanation of the lower glycaemic responses in participants with diabetes. This has been suggested to be tested via comparing those with diabetes during medication withdrawal and prescription [11].

## 4. Current and Critical Perspectives and Future Research

Current rhetoric surrounding hydration is such that the addition of fluids, primarily water, in the diet is good for health. This brief narrative review focuses on glucoregulatory health, which has links to overall metabolic health and disease (such as T2D). It should be noted that the perspectives herein are solely related to the addition of fluids to improve hydration status (i.e., not substitution of energy containing beverages) and the impact on glucoregulatory health; other outcomes or contexts may be altered differentially and therefore may not be applicable to the perspectives presented. For example, growing evidence suggests higher fluid intake to reduce the concentration of urine may aid in kidney health [37] or reduce the risk of urinary tract infection recurrence [38], and that hydration status may influence endocrine responses to exercise [39].

This section therefore aims to: Critically discuss the differences found between participants who are healthy and who have diabetes; critically evaluate the role of the HPA axis in hypohydration-mediated AVP secretion; clarify the purpose of interventions that claim to manipulate hydration status (e.g., reduce urine concentration versus increase body water); and provide suggestions for future research directions, including methods to uncouple the effects of manipulating hydration status (i.e., body water) and circulating AVP concentrations. As it stands, there appears to be limited evidence that hydration status directly alters glucoregulation, particularly in healthy adults. Replicating the current limited research should therefore be a priority. In those with diabetes, the evidence is clearer (though still only two studies [25,27]) but is likely an artefact of glucosuria after euhydration since such effects have only been testing during medication withdrawal. Certainly the glucosuria hypothesis needs to be further examined.

As previously mentioned, direct comparisons between healthy participants and those with diabetes should be made with caution. Of particular interest is the differential postprandial cortisol response found when comparing healthy participants [11] to those with diabetes during medication withdrawal [25,27]. Mechanisms for this interaction are as yet poorly understood but they are unlikely mediated by AVP or the RAAS. The reason these two mechanisms are unlikely is because: (i) The RAAS was not different between hypohydration and euhydration in participants with T2D [27]; (ii) AVP concentrations (measured by copeptin) appear to remain elevated at roughly a constant magnitude throughout an OGTT [11], though it is unknown if this is the case in diabetes. Thus, this change in cortisol is more likely to be mediated via other pathways and may be part of a complex interaction related to medication withdrawal and perhaps nutritional status.

A key underlying theory is that AVP acts along the stress response; thus if an individual is hypohydrated during stress, higher AVP will result in higher ACTH (due to AVP potentiating the effects of CRH). Ergo, in theory, maintaining low circulating AVP concentrations would result in lower CRH cleavage into ACTH, mitigating cortisol-mediated hepatic glucose output (Figure 2). This pathway was determined primarily from theory and animal models. We, however, found no evidence that hydration status induced a difference in ACTH or cortisol concentrations, despite meaningful elevations in copeptin concentrations, even under physical stress (i.e., muscle biopsies) [11]. In healthy adults, a large degree (~5% body mass loss) might be needed to induce an elevation in fasting circulating cortisol concentrations [39], which is not representative of daily fluctuations in the water balance. At a more typical level of body mass loss (~2.5%) no differences were found in cortisol concentrations [39], in accordance with our data [11]. Such differences between 2.5% and 5% hypohydration may help explain why the high dose AVP infusion caused higher plasma glucose compared to the low dose in previous work (18). Thus, if AVP does potentiate the effects of CRF, this is unlikely via hydration-mediated AVP changes, at least during every day fluctuations in water balance.

If we therefore examine the totality of evidence critically, one of the conclusions that could be made is that AVP is maybe only partially implicated in glucoregulation, however this is perhaps independent of hydration status. In other words, our hypothesis is that the physiological effects of hypohydration-induced AVP secretion counter-regulate AVP-induced hyperglycaemia, or the effects of increased AVP from other (non-hydration related) causes interact to cause hyperglycaemia, similar to the potential interaction between nutritional status and cortisol secretion found in those with T2D [27]. Alternatively, it could be that there is a residual factor, as yet unknown, that influences both AVP and glucoregulation that is context-specific, thereby explaining why there appears to be no direct effect of hypohydration on glucoregulation, whereas there does appear to be an effect of AVP infusion. Thus, we propose that there are potentially differences in the physiological responses to hydration status that alter glucoregulation, according to whether the response is exogenous (e.g., infusion of AVP), or endogenous (e.g., restricting fluid to raise circulating AVP).

Therefore, we hypothesise that manipulating hydration status in order to reduce AVP is likely to have minimal, if any effect, on glucoregulation, at least in healthy populations. This has somewhat been demonstrated in the aforementioned one week water intervention (+3 L∙day^−1^ added to habitual intake) [10]. Responders to the intervention had a reduction in fasting plasma glucagon concentrations, but not glucose or insulin concentrations, concordant with the increase in glucagon concentrations found during AVP infusion [22]. Such results add credence to the idea that reducing AVP (measured by copeptin) fails to alter glucoregulation in otherwise healthy individuals, despite increasing glucagon. This perhaps suggests that another counter-regulatory process is occurring to mitigate the hyperglycaemic effects of glucagon. Considering studies investigating those with T1D [25], healthy adults prescribed high water intake [10], and AVP infusion [22], researchers should ensure that glucagon is measured so the effects on primary glucoregulatory hormones can be captured. It is unclear mechanistically why glucagon increased in these studies but did not always result in higher plasma glucose concentrations.

Our hypothesis is specific to endogenous AVP production mediated by non-compartment-specific hypohydration. It has been demonstrated that AVP infusion increases glucose concentrations without increasing serum osmolality (i.e., without necessarily altering hydration status *per se*) [8]. Although there has been no replication work to confirm these findings, some studies have shown an increase in glycaemia when infusing hypertonic (2–5%) saline (which will likely raise AVP concentrations) (12 h hypertonic saline at 1 mL∙kg∙h^−1^ followed by 3 h at 200 mL∙kg∙h^−1^ [23]; and 2 h hypertonic saline infusion at 0.1 mL∙kg∙h^−1^ [24]). In the AVP infusion study by Spruce et al. [22] no effect on blood glucose concentrations was observed after saline infusion (which did not alter AVP concentrations), though details regarding the properties of the infused saline were not given; thus as no effect of infusion was found, it is likely they used isotonic saline, concordant with the control groups in the studies infusing hypertonic saline [23,24]. It is unclear if AVP was altered in these other saline infusion studies [23,24].

Discordance between the AVP infusion [22] and hypohydration-induced AVP elevations [11] may also be explained by nutritional status, i.e., fasted AVP infusion resulted in greater hepatic glucose output (but no change in glucose disposal), whereas in a postprandial state, glucose disposal is more pertinent. This perhaps suggests that AVP acts specifically to increase hepatic glucose metabolism, which may be more directly influenced by infusion studies that create intracellular dehydration. Such intracellular dehydration, particularly in hepatocytes, has been shown to increase glucagon secretion and is thus implicated in glucoregulation [40,41] and could help explain the aforementioned higher glucagon concentrations.

Taken together, these findings may point towards AVP being the main factor in glucoregulation, rather than increased serum osmolality having independent effects (Figure 3). Alternatively or additionally, such studies suggest that both (a) exogenous AVP, and (b) compartmental water distribution changes can result in increased glycaemia. Whilst these are important mechanistic insights, they are not necessarily valid for every day fluctuations in water balance in humans. Further, they may represent a physiological condition present in some people due to other factors such as genetic variations (i.e., not hypohydration).

A counter argument to our hypothesis may be differences related to genetic influences [32,33]. Although these studies highlight genetic variation relating to AVP (and therefore by inference water balance physiology) are correlated to poorer glucoregulatory and metabolic health outcomes, they lack causality. Specifically, such genetic variations may be collinear with other genetic variations that are detrimental to health. If, for the sake of argument, we assume the association is causal, such findings may mean that genetic variation in water balance physiology can cause increased risk of cardiometabolic disease. However, this does not automatically mean that altering water balance behaviours (i.e., increasing fluid intake) will reduce this genetically determined disease risk. This of course should be further investigated as perhaps targeting people with certain variants could increase the likelihood that a water intervention might be efficacious (and in the case of the previously discussed water-prescription study by Enhorning et al. [10], may explain some variation in the responders and non-responders).

Further considerations should also be taken into account in future research. Firstly, the epidemiology investigating water intake and glucoregulatory health is at least suggestive of sex differences. Considering that much physiology research is based on men (e.g., [27]), or when women are included they are in their (estimated) follicular phase, post-menopausal, or taking hormonal contraceptives (e.g., [11]), studies investigating water balance and health outcomes during the luteal phase of the menstrual cycle would be of mechanistic interest. The osmolality set point for AVP secretion changes throughout the menstrual cycle [31] as does carbohydrate and fat oxidation [42] and understanding how these fluctuations influence health would aid in our mechanistic understanding of whether and how AVP influences glucoregulatory health.

Secondly, pre-trial control of known confounding factors should be emphasised. Our recent study included four days of pre-trial diet, activity, and fluid intake standardisation using weighed food diaries and combined accelerometry and heart rate monitors [11]. Such control to our knowledge has not been utilised in previous hydration and health related research. To demonstrate why this may be important, comparatively, our pilot work used 24 h of diet and activity standardisation in the form of verbal confirmation [26]. The stark differences in the results from these studies may indicate that lack of pre-trial standardisation at least partially contributed to divergences in glucoregulation during the OGTT.

Thirdly, another explanation as to why there have been conflicting findings, may be the use of venous versus arterialised-venous blood. In our pilot work, we used venous blood and found a large difference in the blood glucose response between hydration states [26]. Contrarily, in our follow-up study, we used arterialised-venous blood [11]. As arterialised-venous blood more closely represents the glucose concentration that cells are exposed to, whereas venous blood more closely represents the glucose the cells have not taken up, it is reasonable to suspect that this may (at least in part) explain the differences in our findings [43]. Whilst this is a possibility that warrants further investigation, it is worth noting the use of arterialised-venous blood during AVP infusion [22], which still demonstrated a difference in glycaemia. This of course may also be due to the use of exogenous AVP infusion versus endogenous AVP via dehydration.

Further adding doubt to the arterialisation theory is that in our study, after the OGTT, we measured multiple facets of appetite, including serum glucose and insulin concentrations from (non-arterialised) venous blood after an *ad libitum* test meal. During this period of testing, plasma copeptin concentrations and serum osmolality remained elevated during hypohydration but blood glucose and insulin concentrations remained remarkably similar to the euhydrated trial arm [44]. Whilst the *ad libitum* nature of the test meal confounds any definitive inferences, energy intake was (on average) approximately equal between the trial arms, and accordingly, there were no differences in serum glucose or insulin concentrations. If arterialisation was a cause of the disparities between studies, it may have been apparent during this period of testing.

A final consideration is the measurement of hydration status and how this relates to the conclusions of studies. There is currently no gold standard measure of hydration, as each method has its strengths and limitations according to the context. Our recent study, to our knowledge, measured hydration status more extensively than any other research investigating hydration and health [11], including body mass, plasma copeptin concentration, urine and serum osmolality, urine specific gravity, peripheral quantitative computer tomography, muscle biopsies, and fluid intake diaries. Whilst this level of measurement is unnecessary for all research, future work should consider the appropriateness of the measures taken, and a clear distinction needs to be made before starting the trial: Is the aim to alter urine concentration, AVP, or body water?

If the aim of the study is to increase urine volume or decrease urine concentration, then measures such 24-h urine volume, urine osmolality, or urine specific gravity are suitable. However, these measures alone do not indicate that hydration status has been altered, though they may be sufficient to infer (with caution) that AVP has been manipulated. Changes in these outcomes simply demonstrates that the body has no (or at least less) need to reabsorb extra fluid, or fluid has been consumed in a way that is conducive to increased/decreased urinary output such as consuming a large bolus of fluid rapidly (e.g., Shafiee et al. [45]).

Equally, measuring AVP (or a marker of) alone does not necessarily infer that body water (hydration status) has been altered. Arginine vasopressin is secreted in order to reduce water losses. Thus at least in the early phase of elevated concentrations, it should be effective at maintaining water balance within the body. Acutely, measuring body mass can be effectively used to determine whether body water has been altered, though this implies energy balance, emphasising the importance of proper pre-measure standardisation of diet and activity. Therefore, it is essential to specify the aim of the study, use the appropriate (combination of) measures, and most importantly to frame conclusions within the correct context (i.e., are the inferences based on reducing urine concentration, circulating AVP, or manipulating body water)?

This raises a wider question regarding how we can improve the measurement of hydration status and accurately assess the contribution of hydration-mediated changes in AVP on glucoregulatory health. We propose two potential pathways that could help uncouple the independent effects of alterations in body water and circulating AVP concentrations: (i) Investigating 3,4-Methylenedioxymethamphetamine (MDMA), and (ii) investigating those with water balance conditions, namely the syndrome of inappropriate antidiuretic hormone (another name for AVP) secretion (SIADH) and diabetes insipidus.

3,4-Methylenedioxymethamphetamine is the psychoactive ingredient in the recreational drug more commonly known as ‘ecstasy’. In terms of hydration, this drug is most fascinating as it gives the symptoms of hypohydration (reduced urine volume, and increased urine osmolality [despite greater fluid ingestion], plasma copeptin concentrations, thirst, desire for fluid, dry mouth, and body temperature), whilst simultaneously causing cell swelling (due to the elevation of AVP resulting in hyponatraemia and greater water retention, as well as the greater fluid ingestion) [36]. Mixed effects have been found for whether plasma osmolality changes from MDMA administration: Two placebo-controlled studies found no effect [46,47], whereas another natural study taking pre- and post-clubbing measured in self-administering participants found a post-MDMA reduction in plasma osmolality [48], with similar results in another placebo-controlled trial though this study did not find an interaction between MDMA and AVP [49]. As studies varied in their levels of control regarding fluid intake, physical activity, and dosage, the effects of MDMA on plasma osmolality need to be clarified in order to fully understand the hydration-AVP-health interactions (described below). For example, if serum osmolality remains unchanged post-MDMA administration, this would provide clear evidence for the effects of AVP on glucoregulation, independent of other hydration-related factors.

Accordingly, MDMA could (a) help improve our understanding of hydration physiology and measurement, and (b) provide a useful model to help assess the role of AVP in glucoregulatory health. With regards to the first point (a), if we were able to find a simple measure of hydration that accurately describes those under the influence of MDMA as hyperhydrated, despite the overwhelming symptoms of hypohydration, we may more effectively be able to assess hydration status. Additionally, such MDMA-induced symptoms have some sex-differences (specifically that copeptin concentration increase more in women than men [47]), further highlighting the usefulness of an MDMA model in understanding the mechanisms surrounding water balance. An unintended consequence of pursuing this line of research may be a reduction in MDMA-related deaths, which are primarily caused by hyponatraemia or hyperhydration.

Regarding the latter point (b), considering MDMA results in both elevated AVP and hyperhydration, if AVP was the cause of hyperglycaemia, this should mean MDMA induces hyperglycaemia. Of course this is very difficult to test for several reasons, such as ethical and legal restrictions, and confounding factors in natural settings such as the temperature and activity patterns of users. As such, there is very limited research. In rats, MDMA administration results in hypoglycaemia [50]. However, in humans administered MDMA in a natural setting, but with no control of food or fluid intake, six out of 21 participants had a non-significant elevation in blood glucose concentrations compared to baseline [51].

It is unclear as to whether MDMA increases circulating cortisol concentrations or not. In a club setting, MDMA increased cortisol [52], whereas in a placebo-controlled setting it did not [46], meaning it is unclear as to whether euglycaemia was maintained in MDMA administered in a natural setting [51] despite changes in cortisol secretion. It may be concluded from these studies that MDMA-induced endogenous AVP secretion is at least not associated with glucose dysregulation. Using an MDMA model of AVP physiology and glucoregulatory outcomes may be useful in aiding our understanding of these complex relationships as it enables us to uncouple the mechanistic effects of hydration status and AVP. Nonetheless, caution should still be taken when making inferences from such results to the general population as MDMA affects a multitude of metabolic and neuronal pathways, which could also be implicated in glucoregulation. However, this does not necessarily detract from the mechanistic understanding that such a model can bring.

Following on from an MDMA model, is investigating effects in those with SIADH. This condition is characterised by low serum osmolality (which can be accompanied by cell swelling and hyponatraemia), highly concentrated urine, and elevated AVP [53]; thus in some ways this condition mimics the water balance effects of MDMA administration [46,48,54] and could also be used as a model to uncouple the effects of body water compared to AVP in glucoregulatory health. Conversely, neurohypophyseal diabetes insipidus is the underproduction of AVP resulting in excessive fluid losses [55] (though there are other forms which can result in elevated AVP, e.g., nephrogenic diabetes insipidus [56]).

We were unable to find data regarding glucoregulatory health in either of these conditions. If the AVP-induced hyperglycaemia model were correct, we would expect to see a greater prevalence of markers of glucose intolerance (e.g., impaired fasting glucose, or higher T2D prevalence) in those with SIADH, and lower prevalence in those with diabetes insipidus. In the absence of clear current data, the relationship between diabetes insipidus and mellitus was previously of interest in the early 1900s; although there were some cases of increased glucosuria with diabetes insipidus, there was no evidence that diabetes mellitus prevalence was different from the general population [57]. Epidemiological work could investigate this relationship further in order to discern whether more causal work is warranted. As with MDMA research, inferences from both SIADH and diabetes insipidus models should be made cautiously when extrapolating to the general population; for example, in SIADH, inappropriate secretin secretion is currently the most likely cause of the condition, rather than problems with the AVP response *per se* [58]. Further, the study of those with diabetes insipidus may help to uncouple the effects of glucosuria and glucoregulation found in those with T1D and T2D during medication withdrawal; this may however, additionally reduce the applicability of this model to the general population. Nonetheless, such models provide useful mechanistic understandings, which can help drive future hypotheses.

Future research should also consider longer-term interventions. Whilst this has ethical implications, there is ample evidence to suggest that some people are chronic low fluid drinkers; understanding the causal implications of this behaviour is essential for public health. Although the acute evidence in healthy adults at least suggests that such an intervention will not cause metabolic harm, reducing some initial ethical concerns, important questions remain to be answered as to whether this lack of harm extends beyond a matter of days. In other words, does chronically high AVP induced by low fluid intake eventually fulfill the HPA axis causing elevated cortisol (notwithstanding other potential mechanisms such as changes in cell volume which may influence glucoregulation [40,41]).

Finally, much of the mechanistic work relating AVP to glucoregulatory health has involved isolating a single mechanism (AVP in the case of the focus of this review, but also hepatocyte volume and more recently, adipocyte AVP receptor expression). Such studies are vitally important for understanding the underlying physiology and generating testable hypotheses. However, the more recent work in humans, which has encompassed the full range of the physiological effects of hypohydration, demonstrates that such models are not necessarily applicable to human fluctuations in water balance. Thus in order to understand the glucoregulatory impacts of AVP, ecologically valid methods should be used in order to make accurate inferences applicable to human health.

## 5. Conclusions

Overall, this narrative review provided a brief account of the history of AVP-glucoregulation related research. Whilst the earlier research gave insight into potential mechanisms backed by observational studies, tightly controlled studies do not at this time appear to support a causal role for hypohydration-induced increases in AVP on glucoregulatory health. Although studies in people with diabetes show euhydration results in lower glycaemia, this is likely due to glucosuria from medication withdrawal; though more research needs to confirm this. Such findings in their totality suggest that AVP may be implicated in glucoregulation particularly at supraphysiological concentrations, but possibly not via every day alterations in non-compartment-specific hydration status, at least in healthy (young) adults. However, research in this field using ecologically valid methods in both healthy participants and those with diabetes is in its infancy. Several ideas for further examining mechanisms and outcomes were also discussed, with an emphasis on replication of current studies.

## Figures and Tables

**Figure 1 nutrients-11-01201-f001:**
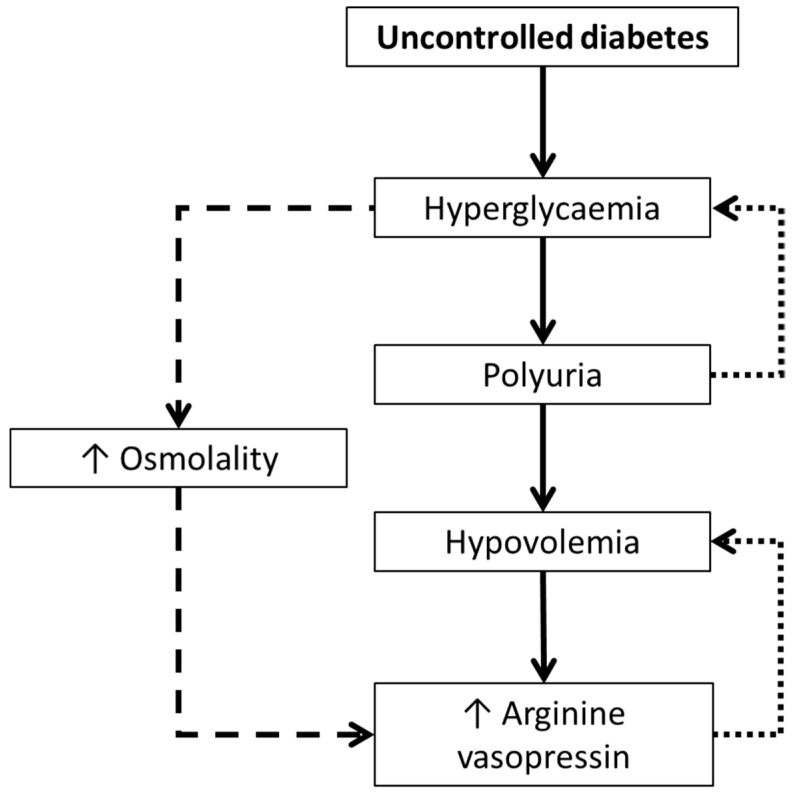
The relationship between diabetes and increased arginine vasopressin, as per the findings of Zerbe et al. (1979) [17]. Dotted lines represent a feedback loop, which aims to maintain homeostasis. Dashed line represents a theoretical pathway whereby hyperglycaemia induces an osmotic stimulus for greater AVP secretion.

**Figure 2 nutrients-11-01201-f002:**
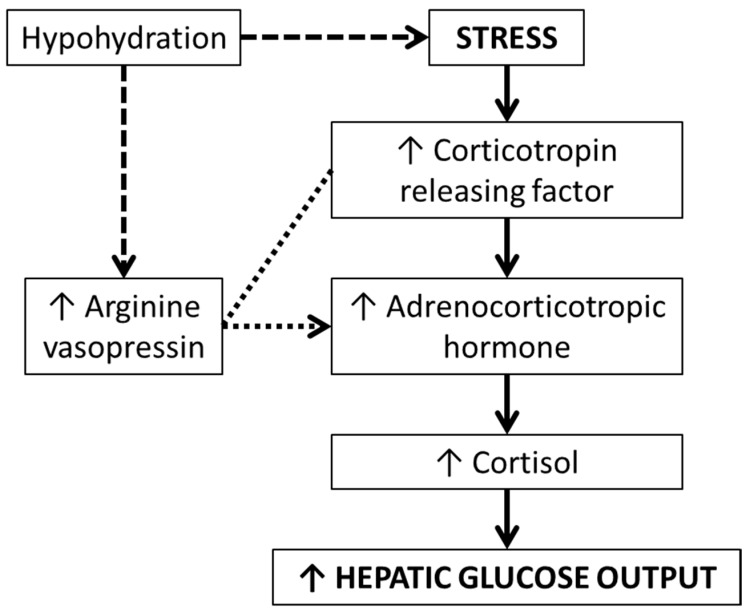
Theoretical relationship between hydration status, arginine vasopressin, and the hypothalamic-pituitary-adrenal axis. Dashed lines represent a mediating relationship (i.e., hydration status directly influences arginine vasopressin, and hydration status may influence stress); dotted lines represent a moderating relationship (i.e., arginine vasopressin determines the propensity of corticotropin releasing factor to be cleaved into adrenocorticotropic hormone).

**Figure 3 nutrients-11-01201-f003:**
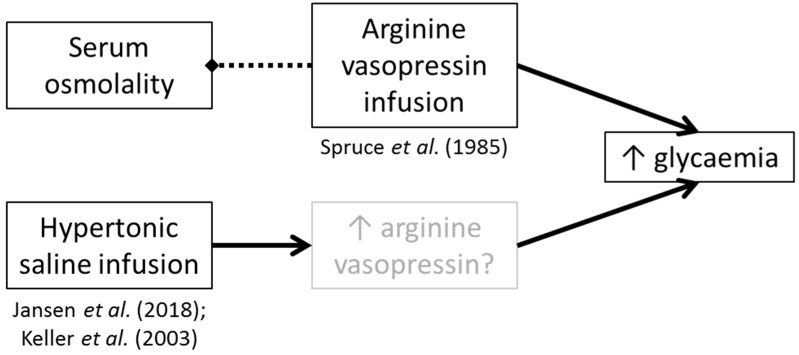
Theoretical relationship between AVP or saline infusion and glucoregulation. Dotted line with diamond arrowhead represents no effect. Greyed out box means unmeasured, but theoretically on the pathway.

**Table 1 nutrients-11-01201-t001:** Literature pertaining to hydration and glucoregulation.

Author, Year	Study Design	Participants	Method/Assessment of HYPO	Level of HYPO Achieved	Glucoregulatory Assessment	Findings
OBSERVATIONAL STUDIES						
Zerbe et al., 1979 [17]	Cross-sectional	*n* = 15 men, 13 women with uncontrolled diabetes	AVP concentrations	N/A	Disease status	↑ AVP associated with poor gluco-regulation
Enhorning et al., 2010 [12]	Cross-sectional and longitudinal	*n* = 1418 healthy, 364 IFG, 205 T2D men, 2284 healthy, 311 IFG, 160 T2D women	Copeptin concentrations	N/A	T2D risk	Cross-sectional: ↑ copeptin associated with ↑ T2D prevalence and IR Longitudinal: ↑ copeptin associated with ↑ T2D (healthy at baseline OR Q1 vs. Q4 2.64; IFG at baseline OR Q1 vs. Q4 3.48)
Roussel et al., 2011 [18]	Longitudinal	*n* = 1707 healthy men, 1908 healthy women	Plain water intake	N/A	Risk of new-onset hyperglycaemia	↑ Water intake associated with ↓ risk of hyperglycaemia (<0.5 vs. <1.0 and >1.0 L/d OR 0.68–0.79)
Pan et al., 2012 [19]	Longitudinal	*n* = 82,902 healthy women	Plain water intake	N/A	T2D risk	× <1 vs. categories up to ≥6 cups/d RR 0.93–1.09
Carroll et al., 2015 [20]	Cross-sectional	*n* = 60 healthy men, 78 healthy women	Plain water intake	N/A	T2D risk score	↑ 1 cup water/d associated with 0.72 ↓ T2D risk score
Carroll et al., 2016 [21]	Cross-sectional	*n* = 456 healthy men, 579 health women	Plain water intake	N/A	HbA1c	Men: ↑ 1 cup water/d associated with ↓ 0.04% HbA1c Women: ↑ 1 cup/d associated with × HbA1c
INFUSION STUDIES						
Spruce et al., 1985 [22]	Randomised crossover trial	*n* = 6 healthy men	IV low then high dose AVP vs. (isotonic?) IV saline	↑ AVP by ≥15 pmol∙L^−1^	Fasted glucose kinetics	↑ Arterialised venous blood glucose concentration (low dose AVP Δ ~0.3, high dose Δ ~0.8 mmol∙L^−1^) × insulin concentration ↑ glucagon concentration (Δ ~41 pg∙L^−1^)
Keller et al., 2003 [23]	Randomised crossover trial	*n* = 10 healthy men	HypoOsm: IV 4 μg desmopressin + 200 mL/h water → IV 4 μg desmopressin + IV mL/h 0.4% saline; vs. HyperOsm: IV 1 mL/kg/h 2% saline → IV 200 mL/h 5%; vs. IsoOsm: *ad libitum* water ingestion	HypoOsm: ↓ P_osm_ ~ 21 mOsm/kg; ↑ body mass (~1.6 kg); ↑ urine output (~1.6 L) HyperOsm: ↑ P_osm_ ~ 13 mOsm/kg; × body mass or urine output IsoOsm: × P_osm_, body mass	Fasting glucose concentrations and hyperinsulinaemic-euglycaemic clamping	↑ Glucose concentration after HyperOsm (5.1 mmol∙L^−1^) vs. HypoOsm (4.7 mmol∙L^−1^) vs. IsoOsm (4.9 mmol∙L^−1^) ↓ Insulin concentration HypoOsm vs. IsoOsm and HyperOsm ↑ Endogenous glucose appearance during HyperOsm vs. IsoOsm and HypoOsm
Jansen et al., 2018 [abstract only] [24]	Randomised crossover trial	*n* = 30 healthy men	HyperOsm: IV 3.0% saline vs. IsoOsm: IV 0.9% saline	HyperOsm: ↑ Posm ~ 18 mOsm/kg IsoOsm: ↑ Posm ~ 3 mOsm/kg	OGTT gluco-regulatory profile	↑ Glucose concentration at 60 (157 vs. 145 mg∙dL^−1^) and 90 (139 vs. 128 mg∙dL^−1^) min HyperOsm vs. IsoOsm
WATER INTAKE MANIPULATION STUDIES						
Burge et al., 2001 [25]	Controlled before-and-after study	*n* = 10 men, 5 women with T1D during insulin withdrawal	Control (euhydrated) phase followed by fluid restriction (750 mL/d) + oral 5 mg metolazone + IV 40–120 mg furosemide	↓ Body mass (4.1%); ↓ body water% (~3%)	Fasted insulin withdrawn gluco-regulatory profile (5 h)	↑ Glucose (6.00 vs. 5.88 mmol∙L^−1^), glucagon (66 vs. 58 ng∙L^−1^), cortisol (497 vs. 384 nmol∙L^−1^) concentrations HYPO vs. control phase × Insulin concentration ↓ Glucosuria (13.9 vs. 27.6 g) HYPO vs. control phase
Carroll et al., 2016 [26]	Pilot randomised crossover trial	*n* = 4 healthy men, 1 healthy woman	HYPO: 45 min sauna + fluid restriction (≤200 mL water between sauna and testing) Control (euhydration): 45 min sauna + ≥ 150% sweat losses in water between sauna and testing	↓ Body mass (~1.3%), ↑ urine osmolality (↑ ~463 mOsm/kg vs. control)	Fasted and OGTT glucose and lactate concentrations	↑ Glucose concentrations at 45 (5.88 vs. 4.74 mmol∙L^−1^) and 60 (4.87 vs. 4.09 mmol∙L^−1^) HYPO vs. control ↑ Glucose iAUC (72.9 vs. 66.6 mmol × 120 min∙L^−1^) HYPO vs. control × Lactate concentration
Johnson et al., 2017 [27]	Randomised crossover trial	*n* = 9 men with T2D during medication withdrawal	HYPO: 24–72 h pre-trial, 1 L/d + medication withdrawal → 24 h pre-trial, 0.5 L water + medication withdrawal Control (euhydration): 72 h pre-trial 3 L/d water + medication withdrawal	↓ Body mass (1.5%), ↑ urine specific gravity (~0.018), urine osmolality (~482 mOsm/kg), P_osm_ (~10 mOsm/kg), serum sodium (~3 mEq/L) HYPO vs. control	Fasted and OGTT gluco-regulatory profile	× Fasted glucose, insulin, cortisol, plasma renin activity, aldosterone concentrations ↑ Postprandial glucose concentration HYPO vs. control (AUC 1822 vs. 1689 mmol∙L^−1^∙min^−1^) × Postprandial insulin, plasma renin activity, or aldosterone concentrations ↓ Postprandial cortisol concentration control vs. HYPO (interaction *p* = 0.017, but no differences between time points)
Enhorning et al., 2019 [10]	Randomised crossover trial	*n* = 9 healthy men, 28 healthy women	(i) Acute 1 L (vs. 10 mL) water intake and copeptin (ii) 1 week 3 L/d added water intake and copeptin vs. control (habitual intake)	N/A	Fasted and OGTT gluco-regulatory profile	Acute: × glucose or insulin concentrations during OGTT but ↓ glucagon concentration 1 week intervention: × fasted glucose, insulin, glucagon concentrations
Carroll et al., 2019 [11]	Randomised crossover trial	n = 8 healthy men, 8 healthy women	HYPO: 1 h heat tent + 3 mL/kg body mass/~34 h water Control (euhydration): 1 h heat tent + 150% sweat losses + 40 mL/kg lean body mass in water	↓ Body mass (1.9%), CSMA (365 mm^2^), muscle water (~11.1 g/kg vs. control), ↑ urine specific gravity (~0.010), urine osmolality (~442 mOsm/kg), serum osmolality (9 mOsm/kg), copeptin (14.32 pmol∙L^−1^) × In above after control condition	Fasted and OGTT gluco-regulatory profile	× Fasted or postprandial glucose, insulin, ACTH, or cortisol concentrations HYPO vs. control

Abbreviations and symbols: Δ, change; ×, no change/no difference; ~, approximately; ↓, decreased; ↑, increased; →, followed by; ACTH, adrenocorticotropic hormone; AUC, area under curve; AVP, arginine vasopressin; CSMA, cross-sectional muscle area; d, day; HbA1c, glycated haemoglobin; HYPO, hypohydration; HyperOsm, hyperosmolality trial arm; HypoOsm, hypoosmolality trial arm; iAUC, incremental area under the curve; IFG, impaired fasting glucose; IR, insulin resistance; IsoOsm, isoosmolality trial arm; IV, intravenous infusion; OGTT, oral glucose tolerance test; OR, odds ratio; P_osm_, plasma osmolality; Q, quartile; RR, relative risk; T1D, type 1 diabetes; T2D, type 2 diabetes; vs., versus.

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
