# Peer review of "Hydration, Arginine Vasopressin, and Glucoregulatory Health in Humans: A Critical Perspective"

_nutrients, 2019, doi:10.3390/nu11061201_

Round 1
Reviewer 1 Report
1) Very interesting topic and approach of the bibliographic review. I suggest adding a paragraph about the effects of dehydration on cortisol, therefore on the regulation of glucose at rest and during exercise.
1. Castro-Sepulveda M, Ramirez-Campillo R, Abad-Colil F, Monk C, Peñailillo L, Cancino J, Zbinden-Foncea H. Basal Mild Dehydration Increase Salivary Cortisol After a Friendly Match in Young Elite Soccer Players. Front Physiol. 2018 Sep 26; 9: 1347.
2. López-Samanes Á, G Pallarés J, Pérez-López A, Mora-Rodríguez R, Ortega JF. Hormonal and neuromuscular responses during a match in male professional tennis players. PLoS One. 2018 Apr 6; 13 (4): e0195242
2) Add a summary table with studies in humans with the effects of dehydration on arginine vasopressin and glucose levels.
Author Response
[Reviewer comment] 1) Very interesting topic and approach of the bibliographic review. I suggest adding a paragraph about the effects of dehydration on cortisol, therefore on the regulation of glucose at rest and during exercise.
1. Castro-Sepulveda M, Ramirez-Campillo R, Abad-Colil F, Monk C, Peñailillo L, Cancino J, Zbinden-Foncea H. Basal Mild Dehydration Increase Salivary Cortisol After a Friendly Match in Young Elite Soccer Players. Front Physiol. 2018 Sep 26; 9: 1347.
2. López-Samanes Á, G Pallarés J, Pérez-López A, Mora-Rodríguez R, Ortega JF. Hormonal and neuromuscular responses during a match in male professional tennis players. PLoS One. 2018 Apr 6; 13 (4): e0195242
[Author response] We felt we had adequately described the impact of hypohydration on cortisol and the surrounding theory regarding the HPA axis (lines 129-140, and referred to throughout at relevant points). Considering the multitude of factors that change with exercise, we opted to only focus on resting gluco-regulation (as stated on line 298). However, we note that in one of the references provided (Castro-Sepulveda et al.) there were rested data for salivary cortisol based on hydration status (assessed via USG) which showed no difference, supporting our hypothesis. However, considering the natural study design—which included no control of food intake prior to cortisol measures—and that this was in adolescents—who may have differential cortisol responses compared to adults (e.g. https://www.ncbi.nlm.nih.gov/pmc/articles/PMC4274618/)—we felt there were too many confounding factors to include this as a fair comparable piece of research to support our theory.
[Reviewer comment] 2) Add a summary table with studies in humans with the effects of dehydration on arginine vasopressin and glucose levels.
[Author response] Thank you for this suggestion. We have included a table as per your suggestion.
Reviewer 2 Report
The submitted manuscript is very original and provides interesting ideas for future research. However, in different sections I consider it difficult to follow and quite confusing. Moreover, despite being a narrative review, I consider that the number of references is scarce.
Author Response
[Reviewer comment] The submitted manuscript is very original and provides interesting ideas for future research. However, in different sections I consider it difficult to follow and quite confusing. Moreover, despite being a narrative review, I consider that the number of references is scarce
[Author response] Thank you for your overall positive impression. We appreciate this is a highly theoretical piece which can be hard to follow; therefore we have aimed to improve the clarity of the manuscript to avoid confusion, for example, in the paragraph lines 337-355:
“If we therefore examine the totality of evidence critically, one of the conclusions that could be made is that AVP is maybe only partially implicated in gluco-regulation, however this is perhaps independent of hydration status. In other words, our hypothesis is that the physiological effects of hypohydration-induced AVP secretion counter-regulate AVP-induced hyperglycaemia, or the effects of increased AVP from other (non-hydration related) causes interact to cause hyperglycaemia, similar to the potential interaction between nutritional status and cortisol secretion found in those with T2D[24]. Alternatively, it could be that there is a residual factor, as yet unknown, that influences both AVP and gluco-regulation that is context-specific, thereby explaining why there appears to be no direct effect of hypohydration on gluco-regulation, whereas there does appear to be an effect of AVP infusion.“
We have added:
“Thus, we propose there are potentially differences in the physiological responses to hydration status that alter gluco-regulation, according to whether the response is exogenous (e.g. infusion of AVP), or endogenous (e.g. restricting fluid to raise circulating AVP).” (lines 355-358) to help clarify the idea presented in the paragraph.
We appreciated the opportunity to be invited to write this paper; considering the scarcity of literature in this field, we hope you can appreciate the low number of references is an artefact of the low number of studies conducted in the field. As there is a paucity of evidence, we felt this would be an exciting opportunity to outline ideas and testable hypotheses to help the field progress.
Reviewer 3 Report
The paper is well written and can be acceptable in the current form with minor English changes.
Author Response
[Reviewer comment] The paper is well written and can be acceptable in the current form with minor English changes.
[Author response] Thank you for your review. We have checked the manuscript to ensure the English is acceptable and to improve clarity (also as per Reviewer 2’s feedback).
Additional author edits: We have edited Figure 1; our figure legend had specified a dashed line to show our own theory, but we had initially put a solid line. This has now been rectified.
We have checked for typographical errors and corrected these throughout (see tracked changes in uploaded manuscript)
Round 2
Reviewer 1 Report
Thanks for the response to my comments. The article was very consistent and very good information.
Author Response
[Reviewer comment] Thanks for the response to my comments. The article was very consistent and very good information.
[Author response] Thank you again for reviewing our manuscript.
Reviewer 2 Report
The review is reasonably clear and well written; however, there are certain places where the text needs to be clarified (see specific comments below). I have several comments that the authors may wish to address in order to strengthen the manuscript.
Consider writing the acronym of “hidration status” HS
Lines 53-63 some references are missing
Line 60: can drugs modify AVP concentrations?
Line 66: Remover number 3 or if it is a reference, cite it properly
Lines 68 and 69 and in subsequent paragraphs of the text. Excessive names of authors are included in the text. Removing most of them would make the text easier to understand and follow
Line 79: both types of diabetes?
Lines 76 to 85: some references missing
Line 126: In vivo must be written in italics
Lines 119-130: What about synthetic corticoids? Do they show the same effect?
Lines 140-141: references missing
Line 145-147: How was water intake evaluated in each study? Maybe this could have effects in the differences found…
Line 159-160: Reference missing
Line 172: Knockout must be written in italics
Lines 188-189: Are there any differences between “responders” and “non-responders”?
Line 191: After Spruce et al (1985), remove number 8 or cite the reference in a suitable way
Line 196: I would suggest separating references 6a and b…
Lines 208: consider write the acronym of “type 1 diabetes” as T1D
Lines 236-238: confusing sentence, rewrite please
Line 240: which study you refer in this sentence?
Lines 245-246: was this study performed after the study you cite as reference number 25? It seems confusing in the text
Line 284: “though still only two studies” Please, include references.
Line 292: RAAS, acronym not previously defined in the text.
Line 309: “in accordance with our data26” Reference bad cited.
Line 340: Consider writing “per se” in italics
Lines 415-423 Consider writing “ad libitum” in italics
Lines 425-431: How did you evaluate hydration status in your study? Are there any available validated questionnaire?
From my point of view, a general read through again will be great in order to correct some mistakes!
Authors deeply focused on the relationship between hydration and gluco-regulatory diseases. However, they did not mention anything about drugs used for the treatment of these diseases. Are there any interactions between these drugs and hydration status?
Reference section are doubled numbered

Author Response
[Reviewer comment] The review is reasonably clear and well written; however, there are certain places where the text needs to be clarified (see specific comments below). I have several comments that the authors may wish to address in order to strengthen the manuscript.
[Author response] Thank you, we appreciate your time in highlighting specific areas that were unclear and we have addressed these below, as well as re-reading the manuscript to ensure we feel everything is clear.
[Reviewer comment] Consider writing the acronym of “hidration status” HS
[Author response] As there are already quite a lot of abbreviations in the paper, we feel this abbreviation is not necessary.
[Reviewer comment] Lines 53-63 some references are missing
[Author response] Thank you for highlighting the lack of citations in this paragraph, we have now updated this (lines 60-64).
[Reviewer comment] Line 60: can drugs modify AVP concentrations?
[Author response] Ignoring drugs that are designed to alter AVP, there is some evidence that some drugs can influence AVP secretion – we have added pharmaceuticals to this paragraph with the following reference: doi: https://doi.org/10.1210/er.2009-0009 (line 63).
[Reviewer comment] Line 66: Remover number 3 or if it is a reference, cite it properly
[Author response] This has been edited (line 67).
[Reviewer comment] Lines 68 and 69 and in subsequent paragraphs of the text. Excessive names of authors are included in the text. Removing most of them would make the text easier to understand and follow
[Author response] We have checked the manuscript throughout and removed names as per your suggestion. Thank you for this comment; we feel this has made the manuscript much easier to follow.
[Reviewer comment] Line 79: both types of diabetes?
[Author response] The authors do not specify in their paper, other than diabetes mellitus. We have updated the manuscript to include the word ‘mellitus’ for added clarity since later in the paper we discuss insipidus (line 79).
[Reviewer comment] Lines 76 to 85: some references missing
[Author response] These were the findings of the study by Zerbe et al. which the paragraph is referring to. We have included the Zerbe reference throughout to enhance clarity (lines 83-86).
[Reviewer comment] Line 126: In vivo must be written in italics
[Author response] This has been changed.
[Reviewer comment] Lines 119-130: What about synthetic corticoids? Do they show the same effect?
[Author response] There is evidence that infused (i.e. synthetic) cortisol increases blood glucose concentrations: https://www.ncbi.nlm.nih.gov/pubmed/11724664 though we are not clear as to how this would fit into this paragraph, since the paragraph is discussing the theoretical pathway between AVP, the stress response, and blood glucose rather than an in-depth discussion of the effects of corticoids on gluco-regulation. However, we did notice that it was not clear the relation between the HPA axis and gluco-regulation until the last sentence of the paragraph, so we have added “The HPA axis is implicated in the stress response which can increase hepatic glucose output…” (lines 131-132).
[Reviewer comment] Lines 140-141: references missing
[Author response] The aim of this short paragraph was to introduce the next phase of research in the 2000s, thus adding in all the references cited in the previous section seems unclear. To improve the clarity though, we have included the phrase “as discussed in the previous section” (line 150):
“As discussed in the previous section, the studies conducted in the 1970s and 1980s provided fascinating insights into the role of hydration and AVP in gluco-regulatory health.”
[Reviewer comment] Line 145-147: How was water intake evaluated in each study? Maybe this could have effects in the differences found…
[Author response] Thank you for highlighting this crucial detail. This section has been updated to include that water intake was self-reported in each study using a food frequency questionnaire or diet diary, and added that this may be a cause of the mixed findings (lines 157-168):
“Water intake can be used as a crude proxy for hydration status. Few studies have investigated the relationship between water intake and markers of gluco-regulatory health. In a French cohort of men and women, higher water intake (assessed via self-reported categories of litres of plain water per day) was associated with lower risk of hyperglycaemia(22). This relationship was replicated in a small UK sample of men and women(23) but not in a sample of US female nurses(24) (both of which assessed water intake using a food frequency questionnaire). Following these studies, a representative UK sample also found an inverse relationship between plain water intake (from 4-day unweighed diet diaries) and gluco-regulatory health, though upon further analysis, this association was only found in men(25). This latest study may explain the null relationship found in US nurses(24) due to the sample being exclusively women, compared to the other studies which used a mixed sex sample. Regarding AVP, this could make sense due to fluctuations in the osmolality set-point for AVP secretion during the menstrual cycle(26), which may cloud any associations. Alternatively, the unvalidated methods of fluid intake may not have accurately captured true water intake, explaining the mixed findings.”
[Reviewer comment] Line 159-160: Reference missing
[Author response] Apologies, we have now included the reference (line 173).
[Reviewer comment] Line 172: Knockout must be written in italics
[Author response] We have opted not to italicise this as it does not seem to be the convention.
[Reviewer comment] Lines 188-189: Are there any differences between “responders” and “non-responders”?
[Author response] Unless you require further information, we had included the core differences reported in the original research regarding the differences between responders and non-responders (i.e. water intake, copeptin concentrations, and glucagon concentrations) (lines 199-209).
[Reviewer comment] Line 191: After Spruce et al (1985), remove number 8 or cite the reference in a suitable way
[Author response] We have removed the reference to Spruce and instead opted to describe the study in line with your previous suggestion to reduce the amount of author names in the manuscript (line 206)
[Reviewer comment] Line 196: I would suggest separating references 6a and b…
[Author response] We have done this.
[Reviewer comment] Lines 208: consider write the acronym of “type 1 diabetes” as T1D
[Author response] We have changed “type 1 diabetes” to “T1D” throughout.
[Reviewer comment] Lines 236-238: confusing sentence, rewrite please
[Author response] We have re-written (lines 248-250): “Alternatively, the cortisol trend may be a response to the medication withdrawal, since it is similar (though with a more rapid onset) to the work in insulin-withdrawn participants with T1D(33).”
[Reviewer comment] Line 240: which study you refer in this sentence?
[Author response] We have included Zerbe and Spruce references in this sentence (line 254).
[Reviewer comment] Lines 245-246: was this study performed after the study you cite as reference number 25? It seems confusing in the text
[Author response] We have clarified as follows (line 258-259): “As per the AVP-infusion study by Spruce et al.(18) we also did not find a difference in lactate in our pilot work(35), though it tended to be higher during hypohydration 60 min post-glucose ingestion.”
[Reviewer comment] Line 284: “though still only two studies” Please, include references.
[Author response] We have included these citations (line 311).
[Reviewer comment] Line 292: RAAS, acronym not previously defined in the text.
[Author response] RAAS was defined on line 236.
[Reviewer comment] Line 309: “in accordance with our data26” Reference bad cited.
[Author response] This has been corrected (line 336).
[Reviewer comment] Line 340: Consider writing “per se” in italics
[Author response] This has been corrected throughout.
[Reviewer comment] Lines 415-423 Consider writing “ad libitum” in italics
[Author response] This has been corrected throughout.
[Reviewer comment] Lines 425-431: How did you evaluate hydration status in your study? Are there any available validated questionnaire?
[Author response] We have now included this information (lines 471-472): “Our recent study, to our knowledge, measured hydration status more extensively than any other research investigating hydration and health(11), including body mass, plasma copeptin concentration, urine and serum osmolality, urine specific gravity, peripheral quantitative computer tomography, muscle biopsies, and fluid intake diaries.”
[Reviewer comment] From my point of view, a general read through again will be great in order to correct some mistakes!
[Author response] Your positive comments and input on helping clarify the manuscript are much appreciated!
[Reviewer comment] Authors deeply focused on the relationship between hydration and gluco-regulatory diseases. However, they did not mention anything about drugs used for the treatment of these diseases. Are there any interactions between these drugs and hydration status?
[Author response] Drugs (such as metformin) for T2D can cause lactic acidosis (related to a deterioration in kidney function) and increase glucosuria which may affect hydration (e.g. https://www.ncbi.nlm.nih.gov/pubmed/28152176). Whilst this is an interesting topic, we feel it is a bit beyond the scope of this review. Further, it would add complexity to the theories we have presented, particularly as the two studies investigating hydration status in people with T1D and T2D withdrew participants from their medication. Our review has focused on the relationship between hydration and AVP can alter gluco-regulation, but including extra information regarding drug-hydration interactions seems to deviate from this focus. Where appropriate, we have offered some perspective regarding the effects of withdrawing diabetes medication, e.g. (lines 281-282) “This has been suggested to be tested via comparing those with diabetes during medication withdrawal and prescription(11).”
Additionally, when discussing Zerbe’s work, we have included information regarding when the participants were given medication (lines 85-86): “When treated with insulin (and hydrated), plasma AVP concentration reduced by up to five-fold, along with concomitant reductions in plasma osmolality.”
[Reviewer comment] Reference section are doubled numbered
[Author response] This was automatically done by our referencing software, but we have now corrected this.